# ON THE IMPORTANCE OF SINGLE DIRECTIONS FOR GENERALIZATION

**Ari S. Morcos**[1], **David G.T. Barrett, Neil C. Rabinowitz, & Matthew Botvinick**
DeepMind
London, UK
`{arimorcos,barrettdavid,ncr,botvinick}@google.com`

## ABSTRACT

Despite their ability to memorize large datasets, deep neural networks often achieve good generalization performance. However, the differences between the learned solutions of networks which generalize and those which do not remain unclear. Additionally, the tuning properties of single directions (defined as the activation of a single unit or some linear combination of units in response to some input) have been highlighted, but their importance has not been evaluated. Here, we connect these lines of inquiry to demonstrate that a network's reliance on single directions is a good predictor of its generalization performance, across networks trained on datasets with different fractions of corrupted labels, across ensembles of networks trained on datasets with unmodified labels, across different hyperparameters, and over the course of training. While dropout only regularizes this quantity up to a point, batch normalization implicitly discourages single direction reliance, in part by decreasing the class selectivity of individual units. Finally, we find that class selectivity is a poor predictor of task importance, suggesting not only that networks which generalize well minimize their dependence on individual units by reducing their selectivity, but also that individually selective units may not be necessary for strong network performance.

## 1 INTRODUCTION

Recent work has demonstrated that deep neural networks (DNNs) are capable of memorizing extremely large datasets such as ImageNet (Zhang et al., 2017). Despite this capability, DNNs in practice achieve low generalization error on tasks ranging from image classification (He et al., 2015) to language translation (Wu et al., 2016). These observations raise a key question: why do some networks generalize while others do not?

Answers to these questions have taken a variety of forms. A variety of studies have related generalization performance to the flatness of minima and PAC-Bayes bounds (Hochreiter & Schmidhuber, 1997, Keskar et al., 2017, Neyshabur et al., 2017, Dziugaite & Roy, 2017), though recent work has demonstrated that sharp minima can also generalize (Dinh et al., 2017). Others have focused on the information content stored in network weights (Achille & Soatto, 2017), while still others have demonstrated that stochastic gradient descent itself encourages generalization (Bousquet & Elisseeff, 2002, Smith & Le, 2017, Wilson et al., 2017).

Here, we use ablation analyses to measure the reliance of trained networks on single directions. We define a single direction in activation space as the activation of a single unit or feature map or some linear combination of units in response to some input. We find that networks which memorize the training set are substantially more dependent on single directions than those which do not, and that this difference is preserved even across sets of networks with identical topology trained on identical data, but with different generalization performance. Moreover, we found that as networks begin to overfit, they become more reliant on single directions, suggesting that this metric could be used as a signal for early stopping.

---

[1]Corresponding author: arimorcos@google.com

We also show that networks trained with batch normalization are more robust to cumulative ablations than networks trained without batch normalization and that batch normalization decreases the class selectivity of individual feature maps, suggesting an alternative mechanism by which batch normalization may encourage good generalization performance. Finally, we show that, despite the focus on selective single units in the analysis of DNNs (and in neuroscience; Le et al., 2011, Zhou et al., 2014, Radford et al., 2017, Britten et al., 1992), the class selectivity of single units is a poor predictor of their importance to the network's output.

## 2 Approach

In this study, we will use a set of perturbation analyses to examine the relationship between a network's generalization performance and its reliance upon single directions in activation space. We will then use a neuroscience-inspired measure of class selectivity to compare the selectivity of individual directions across networks with variable generalization performance and examine the relationship between class selectivity and importance.

### 2.1 Summary of models and datasets analyzed

We analyzed three models: a 2-hidden layer MLP trained on MNIST, an 11-layer convolutional network trained on CIFAR-10, and a 50-layer residual network trained on ImageNet. In all experiments, ReLU nonlinearities were applied to all layers but the output. Unless otherwise noted, batch normalization was used for all convolutional networks (Ioffe & Szegedy, 2015). For the ImageNet ResNet, top-5 accuracy was used in all cases.

**Partially corrupted labels** As in Zhang et al. (2017), we used datasets with differing fractions of randomized labels to ensure varying degrees of memorization. To create these datasets, a given fraction of labels was randomly shuffled and assigned to images, such that the distribution of labels was maintained, but any true patterns were broken.

### 2.2 Perturbation analyses

**Ablations** We measured the importance of a single direction to the network's computation by asking how the network's performance degrades once the influence of that direction was removed. To remove a coordinate-aligned single direction , we clamped the activity of that direction to a fixed value (i.e., ablating the direction). Ablations were performed either on single units in MLPs or an entire feature map in convolutional networks. For brevity, we will refer to both of these as 'units.' Critically, all ablations were performed in activation space, rather than weight space.

More generally, to evaluate a network's reliance upon sets of single directions, we asked how the network's performance degrades as the influence of increasing subsets of single directions was removed by clamping them to a fixed value (analogous to removing increasingly large subspaces within activation space). This analysis generates curves of accuracy as a function of the number of directions ablated: the more reliant a network is on low-dimensional activation subspaces, the more quickly the accuracy will drop as single directions are ablated.

Interestingly, we found that clamping the activation of a unit to the empirical mean activation across the training or testing set was more damaging to the network's performance than clamping the activation to zero (see Appendix A.1). We therefore clamped activity to zero for all ablation experiments.

**Addition of noise** As the above analyses perturb units individually, they only measure the influence of coordinate-aligned single directions. To test networks' reliance upon random single directions, we added Gaussian noise to all units with zero mean and progressively increasing variance. To scale the variance appropriately for each unit, the variance of the noise added was normalized by the empirical variance of the unit's activations across the training set.

### 2.3 Quantifying class selectivity

To quantify the class selectivity of individual units, we used a metric inspired by the selectivity indices commonly used in systems neuroscience (De Valois et al., 1982, Britten et al., 1992, Freedman

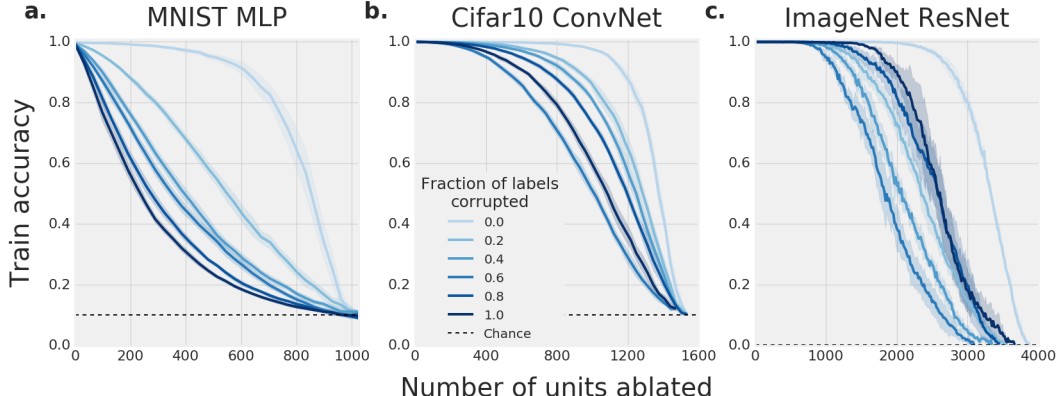

**Figure 1: Memorizing networks are more sensitive to cumulative ablations.** Networks were trained on MNIST (2-hidden layer MLP, **a**), CIFAR-10 (11-layer convolutional network, **b**), and ImageNet (50-layer ResNet, **c**). In **a**, all units in all layers were ablated, while in **b** and **c**, only feature maps in the last three layers were ablated. Error bars represent standard deviation across 10 random orderings of units to ablate.

& Assad, 2006). The class-conditional mean activity was first calculated across the test set, and the selectivity index was then calculated as follows:

$$selectivity = \frac{\mu_{max} - \mu_{-max}}{\mu_{max} + \mu_{-max}} \tag{1}$$

with $\mu_{max}$ representing the highest class-conditional mean activity and $\mu_{-max}$ representing the mean activity across all other classes. For convolutional feature maps, activity was first averaged across all elements of the feature map. This metric varies from 0 to 1, with 0 meaning that a unit's average activity was identical for all classes, and 1 meaning that a unit was only active for inputs of a single class.

We note that this metric is not a perfect measure of information content in single units; for example, a unit with a little information about every class would have a low class selectivity index. However, it does measure the discriminability of classes along a given direction. The selectivity index also identifies units with the same class tuning properties which have been highlighted in the analysis of DNNs (Le et al., 2011, Zeiler & Fergus, 2014, Coates et al., 2012, Zhou et al., 2014, Radford et al., 2017). However, in addition to class selectivity, we replicate all of our results using mutual information, which, in contrast to class selectivity, should highlight units with information about multiple classes, and we find qualitively similar outcomes (Appendix A.5). We also note that while a class can be viewed as a highly abstract feature, implying that our results may generalize to feature selectivity, we do not examine feature selectivity in this work.

## 3 EXPERIMENTS

### 3.1 GENERALIZATION

Here, we provide a rough intuition for why a network's reliance upon single directions might be related to generalization performance. Consider two networks trained on a large, labeled dataset with some underlying structure. One of the networks simply memorizes the labels for each input example and will, by definition, generalize poorly ('memorizing network') while the other learns the structure present in the data and generalizes well ('structure-finding network'). The minimal description length of the model should be larger for the memorizing network than for the structure-finding network. As a result, the memorizing network should use more of its capacity than the structure-finding network, and by extension, more single directions. Therefore, if a random single direction is perturbed, the probability that this perturbation will interfere with the representation of the data should be higher for the memorizing network than for the structure-finding network[2].

---

[2]Assuming that the memorizing network uses a non-negligible fraction of its capacity.

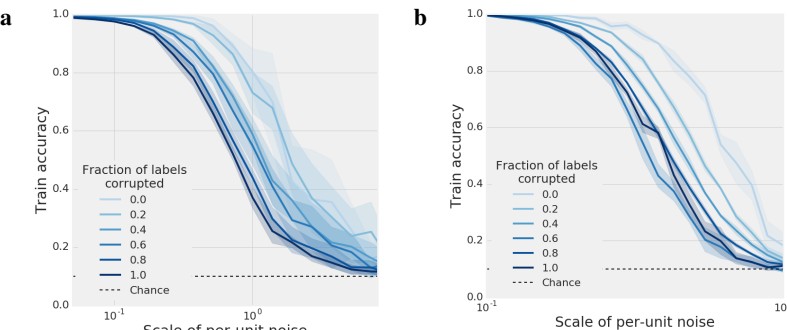

**Figure 2: Memorizing networks are more sensitive to random noise.** Networks were trained on MNIST (2-hidden layer MLP, **a**), and CIFAR-10 (11-layer convolutional network, **b**). Noise was scaled by the empirical variance of each unit on the training set. Error bars represent standard deviation across 10 runs. X-axis is on a log scale.

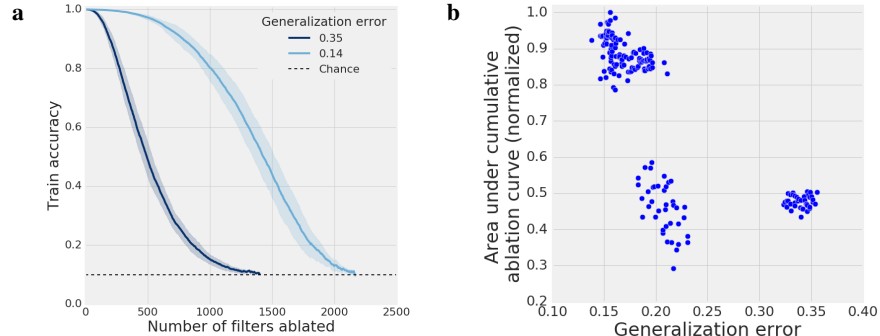

**Figure 3: Networks which generalize poorly are more reliant on single directions.** 200 networks with identical topology were trained on unmodified CIFAR-10. **a**, Cumulative ablation curves for the best and worst 5 networks by generalization error. Error bars represent standard deviation across 5 models and 10 random orderings of feature maps per model. **b**, Area under cumulative ablation curve (normalized) as a function of generalization error.

To test whether memorization leads to greater reliance on single directions, we trained a variety of network types on datasets with differing fractions of randomized labels and evaluated their performance as progressively larger fractions of units were ablated (see Sections 2.2 and 2.1). By definition, these curves must begin at the network's training accuracy (approximately 1 for all networks tested) and fall to chance levels when all directions have been ablated. To rule out variance due to the specific order of unit ablation, all experiments were performed with mutliple random ablation orderings of units. As many of the models were trained on datasets with corrupted labels and, by definition, cannot generalize, training accuracy was used to evaluate model performance. Consistent with our intuition, we found that networks trained on varying fractions of corrupted labels were significantly more sensitive to cumulative ablations than those trained on datasets comprised of true labels, though curves were not always perfectly ordered by the fraction of corrupted labels (Fig. 1).

We next asked whether this effect was present if networks were perturbed along random bases. To test this, we added noise to each unit (see Section 2.2). Again, we found that networks trained on corrupted labels were substantially and consistently more sensitive to noise added along random bases than those trained on true labels (Fig. 2).

The above results apply to networks which are forced to memorize at least a portion of the training set – there is no other way to solve the task. However, it is unclear whether these results would apply to networks trained on uncorrupted data. In other words, do the solutions found by networks with the same topology and data, but different generalization performance exhibit differing reliance upon single directions? To test this, we trained 200 networks on CIFAR-10, and evaluated their generalization error and reliance on single directions. All networks had the same topology and were

trained on the same dataset (unmodified CIFAR-10). Individual networks only differed in their random initialization (drawn from identical distributions), the data order used during training, and their learning rate[3]. We found that the 5 networks with the best generalization performance were more robust to the ablation of single directions than the 5 networks with the worst generalization performance (Fig. 3a). To quantify this further, we measured the area under the ablation curve for each of the 200 networks and plotted it as a function of generalization error (Fig. 3b). Interestingly, networks appeared to undergo a discrete regime shift in their reliance upon single directions; however, this effect might have been caused by degeneracy in the set of solutions found by the optimization procedure, and we note that there was also a negative correlation present within clusters (e.g., top left cluster). These results demonstrate that the relationship between generalization performance and single direction reliance is not merely a side-effect of training with corrupted labels, but is instead present even among sets networks with identical training data.

## 3.2 RELIANCE ON SINGLE DIRECTIONS AS A SIGNAL FOR MODEL SELECTION

This relationship raises an intriguing question: can single direction reliance be used to estimate generalization performance without the need for a held-out test set? And if so, might it be used as a signal for early stopping or hyperpameter selection? As a proof-of-principle experiment for early stopping, we trained an MLP on MNIST and measured the area under the cumulative ablation curve (AUC) over the course of training along with the train and test loss. Interestingly, we found that the point in training at which the AUC began to drop was the same point that the train and test loss started to diverge (Fig. 4a). Furthermore, we found that AUC and test loss were negatively correlated (Spearman's correlation: -0.728; Fig. 4b).

As a proof-of-principle experiment for hyperparameter selection, we trained 192 CIFAR-10 models with different hyperparemeter settings (96 hyperparameters with 2 repeats each; see Appendix A.2). We found that AUC and test accuracy were highly correlated (Spearman's correlation: 0.914; Fig. 4c), and by performing random subselections of 48 hyperparameter settings, AUC selected one of the top 1, 5, and 10 settings 13%, 83%, and 98% of the time, respectively, with an average difference in test accuracy between the best model selected by AUC and the optimal model of only $1 \pm 1.1\%$ (mean $\pm$ std). These results suggest that single direction reliance may serve as a good proxy for hyperparameter selection and early stopping, but further work will be necessary to evaluate whether these results hold in more complicated datasets.

## 3.3 RELATIONSHIP TO DROPOUT AND BATCH NORMALIZATION

**Dropout** Our experiments are reminiscent of using dropout at training time, and upon first inspection, dropout may appear to discourage networks' reliance on single directions (Srivastava et al., 2014). However, while dropout encourages networks to be robust to cumulative ablations up until the dropout fraction used in training, it should not discourage reliance on single directions past that point. Given enough capacity, a memorizing network could effectively guard against dropout by merely copying the information stored in a given direction to several other directions. However, the network will only be encouraged to make the minimum number of copies necessary to guard against the dropout fraction used in training, and no more. In such a case, the network would be robust to dropout so long as all redundant directions were not simultaneously removed, yet still be highly reliant on single directions past the dropout fraction used in training.

To test whether this intuition holds, we trained MLPs on MNIST with dropout probabilities $\in \{0.1, 0.2, 0.3\}$ on both corrupted and unmodified labels. Consistent with the observation in Arpit et al. (2017), we found that networks with dropout trained on randomized labels required more epochs to converge and converged to worse solutions at higher dropout probabilities, suggesting that dropout does indeed discourage memorization. However, while networks trained on both corrupted and unmodified labels exhibited minimal loss in training accuracy as single directions were removed up to the dropout fraction used in training, past this point, networks trained on randomized labels were much more sensitive to cumulative ablations than those trained on unmodified labels (Fig. 5a). Interestingly, networks trained on unmodified labels with different dropout fractions were all similarly robust to cumulative ablations. These results suggest that while dropout may serve as an

---

[3]The learning rate was varied to ensure diverse generalization performance.

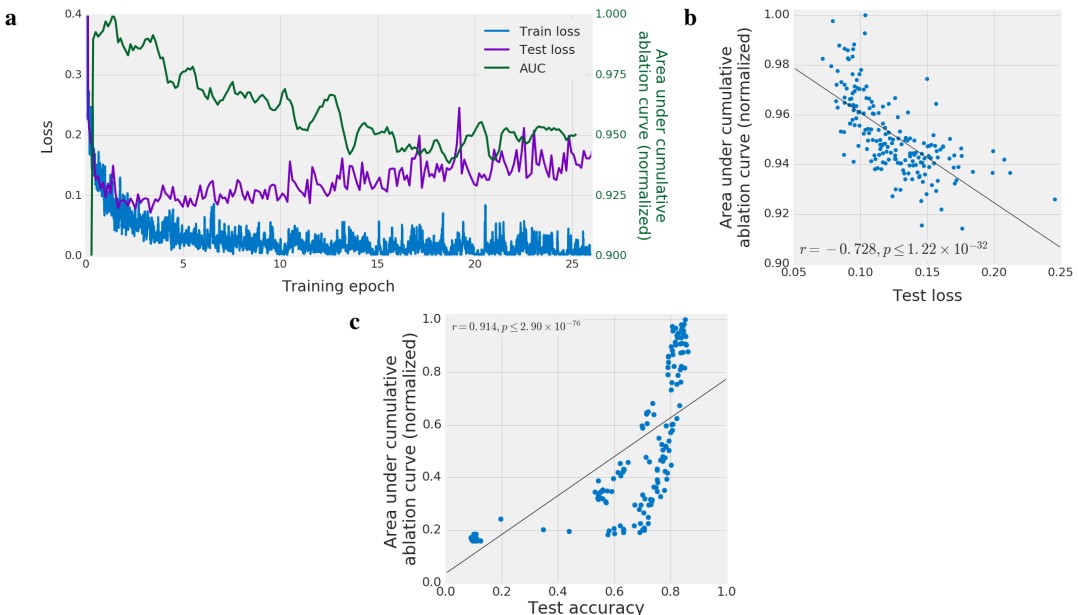

**Figure 4: Single direction reliance as a signal for hyperparameter selection and early stopping.**
**a,** Train (blue) and test (purple) loss, along with the normalized area under the cumulative ablation curve (AUC; green) over the course of training for an MNIST MLP. Loss y-axis has been cropped to make train/test divergence visible. **b,** AUC and test loss for a CIFAR-10 ConvNet are negatively correlated over the course of training. **c,** AUC and test accuracy are positively corrleated across a hyperparameter sweep (96 hyperparameters with 2 repeats for each).

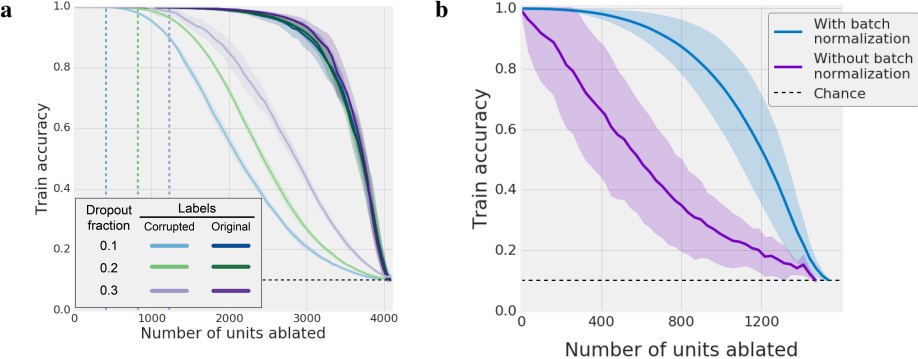

**Figure 5: Impact of regularizers on networks' reliance upon single directions. a,** Cumulative ablation curves for MLPs trained on unmodified and fully corrupted MNIST with dropout fractions $\in \{0.1, 0.2, 0.3\}$. Colored dashed lines indicate number of units ablated equivalent to the dropout fraction used in training. Note that curves for networks trained on corrupted MNIST begin to drop soon past the dropout fraction with which they were trained. **b,** Cumulative ablation curves for networks trained on CIFAR-10 with and without batch normalization. Error bars represent standard deviation across 4 model instances and 10 random orderings of feature maps per model.

effective regularizer to prevent memorization of randomized labels, it does not prevent over-reliance on single directions past the dropout fraction used in training.

**Batch normalization** In contrast to dropout, batch normalization does appear to discourage reliance upon single directions. To test this, we trained convolutional networks on CIFAR-10 with and without batch normalization and measured their robustness to cumulative ablation of single directions. Networks trained with batch normalization were consistently and substantially more robust to these ablations than those trained without batch normalization (Fig. 5b). This result suggests that in

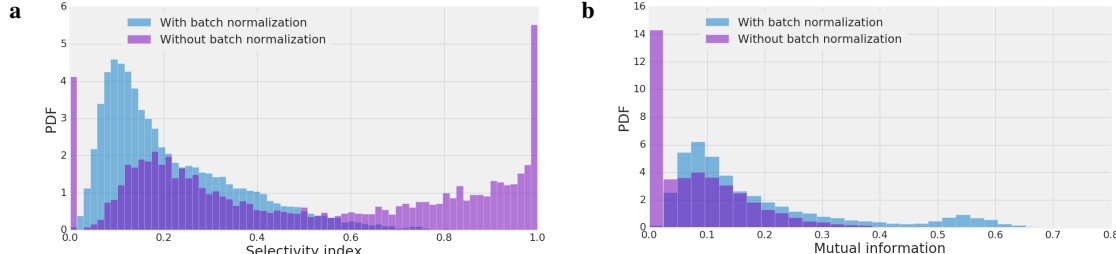

**Figure 6: Batch normalization decreases class selectivity and increases mutual information.** Distributions of class selectivity (**a**) and mutual information (**b**) for networks trained with (blue) and without batch normalization (purple). Each distribution comprises 4 model instances trained on uncorrupted CIFAR-10.

addition to reducing covariate shift, as has been proposed previously (Ioffe & Szegedy, 2015), batch normalization also implicitly discourages reliance upon single directions.

### 3.4 RELATIONSHIP BETWEEN CLASS SELECTIVITY AND IMPORTANCE

Our results thus far suggest that networks which are less reliant on single directions exhibit better generalization performance. This result may appear counter-intuitive in light of extensive past work in both neuroscience and deep learning which highlights single units or feature maps which are selective for particular features or classes (Le et al., 2011, Zeiler & Fergus, 2014, Coates et al., 2012, Zhou et al., 2014, Radford et al., 2017). Here, we will test whether the class selectivity of single directions is related to the importance of these directions to the network's output.

First, we asked whether batch normalization, which we found to discourage reliance on single directions, also influences the distribution of information about class across single directions. We used the selectivity index described above (see Section 2.3) to quantify the discriminability between classes based on the activations of single feature maps across networks trained with and without batch normalization. Interestingly, we found that while networks trained without batch normalization exhibited a large fraction of feature maps with high class selectivity[4], the class selectivity of feature maps in networks trained with batch normalization was substantially lower (Fig. 6a). In contrast, we found that batch normalization increases the mutual information present in feature maps (Fig. 6b). These results suggest that batch normalization actually *discourages* the presence of feature maps with concentrated class information and rather *encourages* the presence of feature maps with information about multiple classes, raising the question of whether or not such highly selective feature maps are actually beneficial.

We next asked whether the class selectivity of a given unit was predictive of the impact on the network's loss of ablating said unit. Since these experiments were performed on networks trained on unmodified labels, test loss was used to measure network impact. For MLPs trained on MNIST, we found that there was a slight, but minor correlation (Spearman's correlation: 0.095) between a unit's class selectivity and the impact of its ablation, and that many highly selective units had minimal impact when ablated (Fig. 7a). By analyzing convolutional networks trained on CIFAR-10 and ImageNet, we again found that, across layers, the ablation of highly selective feature maps was no more impactful than the ablation of non-selective feature maps (Figs. 7b and 7d). In fact, in the CIFAR-10 networks, there was actually a negative correlation between class selectivity and feature map importance (Spearman's correlation: -0.428, Fig. 7b). To test whether this relationship was depth-dependent, we calculated the correlation between class selectivity and importance separately for each layer, and found that the vast majority of the negative correlation was driven by early layers, while later layers exhibited no relationship between class selectivity and importance (Figs. 7c and 7e). Interestingly, in all three networks, ablations in early layers were more impactful than ablations in later layers, consistent with theoretical observations (Raghu et al., 2016). Additionally, we performed all of the above experiments with mutual information in place of class selectivity, and found qualitatively similar results (Appendix A.5).

---

[4]And dead feature maps. Feature maps with no activity would have a selectivity index of 0.

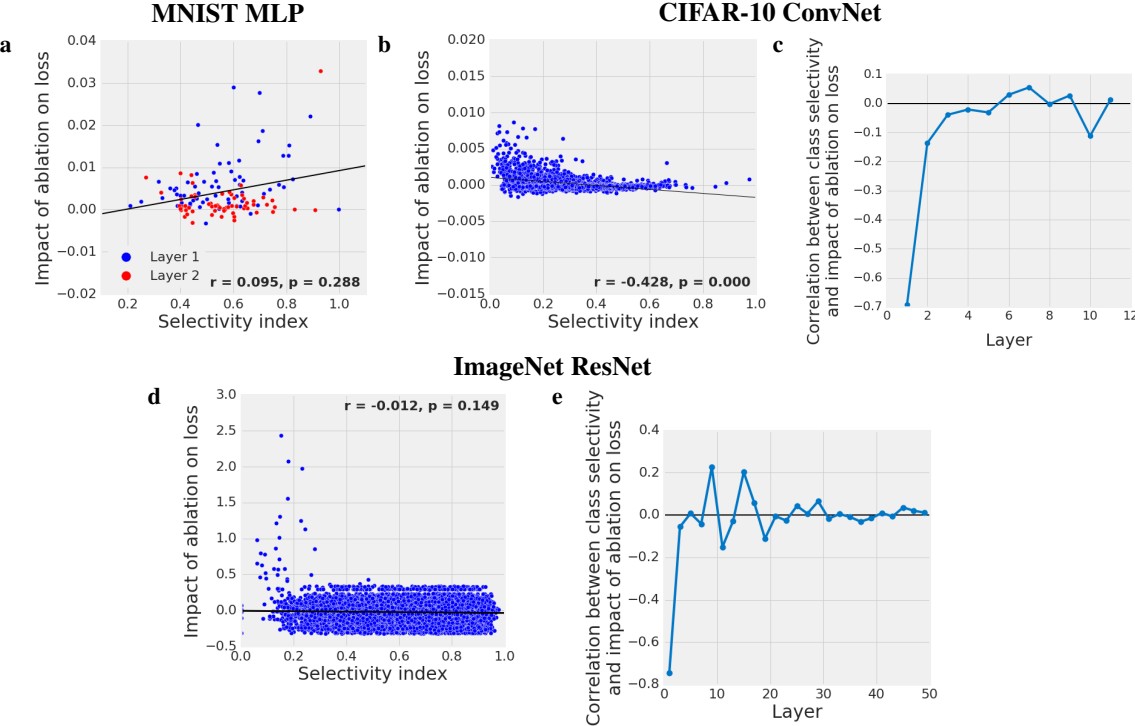

Figure 7: **Selective and non-selective directions are similarly important.** Impact of ablation as a function of class selectivity for MNIST MLP (**a**), CIFAR-10 convolutional network (**b-c**), and ImageNet ResNet (**d-e**). **c** and **e** show regression lines for each layer separately.

As a final test, we compared the class selectivity to the $L^1$-norm of the filter weights, a metric which has been found to be a successful predictor of feature map importance in the model pruning literature (Li et al., 2017). Consistent with our previous observations, we found that class selectivity was largely unrelated to the $L^1$-norm of the filter weights, and if anything, the two were negatively correlated (Fig. A3, see Appendix A.4 for details). Taken together, these results suggest that class selectivity is not a good predictor of importance, and imply that class selectivity may actually be detrimental to network performance. Further work will be necessary to examine whether class and/or feature selectivity is harmful or helpful to network performance.

## 4 RELATED WORK

Much of this work was directly inspired by Zhang et al. (2017), and we replicate their results using partially corrupted labels on CIFAR-10 and ImageNet. By demonstrating that memorizing networks are more reliant on single directions, we also provide an answer to one of the questions they posed: is there an empirical difference between networks which memorize and those which generalize?

Our work is also related to work linking generalization and the sharpness of minima (Hochreiter & Schmidhuber, 1997, Keskar et al., 2017, Neyshabur et al., 2017). These studies argue that flat minima generalize better than sharp minima (though Dinh et al. (2017) recently found that sharp minima can also generalize well). This is consistent with our work, as flat minima should correspond to solutions in which perturbations along single directions have little impact on the network output.

Another approach to generalization has been to contextualize it in information theory. For example, Achille & Soatto (2017) demonstrated that networks trained on randomized labels store more information in their weights than those trained on unmodfied labels. This notion is also related to Shwartz-Ziv & Tishby (2017), which argues that during training, networks proceed first through a loss minimization phase followed by a compression phase. Here again, our work is consistent, as networks with more information stored in their weights (i.e., less compressed networks) should be more reliant upon single directions than compressed networks.

More recently, Arpit et al. (2017) analyzed a variety of properties of networks trained on partially corrupted labels, relating performance and time-to-convergence to capacity. They also demonstrated that dropout, when properly tuned, can serve as an effective regularizer to prevent memorization. However, we found that while dropout may discourage memorization, it does not discourage reliance on single directions past the dropout probability.

We found that class selectivity is a poor predictor of unit importance. This observation is consistent with a variety of recent studies in neuroscience. In one line of work, the benefits of neural systems which are robust to coordinate-aligned noise have been explored (Barrett et al. (2016), Montijn et al. (2016)). Another set of studies have demonstrated the presence of neurons with multiplexed information about many stimuli and have shown that task information can be decoded with high accuracy from populations of these neurons with low individual class selectivity (Averbeck et al. (2006), Rigotti et al. (2013), Mante et al. (2013), Raposo et al. (2014), Morcos & Harvey (2016), Zylberberg (2017)).

Perturbation analyses have been performed for a variety of purposes. In the model pruning literature, many studies have removed units with the goal of generating smaller models with similar performance (Li et al., 2017, Anwar et al., 2015, Molchanov et al., 2017), and recent work has explored methods for discovering maximally important directions (Raghu et al. (2017)). Recently, Cheney et al. (2017) used cumulative ablations to measure network robustness, though the relationship to generalization was not explored. A variety of studies within deep learning have highlighted single units which are selective for features or classes (Le et al., 2011, Zeiler & Fergus, 2014, Coates et al., 2012, Zhou et al., 2014, Radford et al., 2017, Agrawal et al., 2014). Additionally, Agrawal et al. (2014) analyzed the minimum number of sufficient feature maps (sorted by a measure of selectivity) to achieve a given accuracy. However, none of the above studies has tested the relationship between a unit's class selectivity or information content and its necessity to the network's output.

Bau et al. (2017) have quantified a related metric, concept selectivity, across layers and networks, finding that units get more concept-selective with depth, which is consistent with our own observations regarding class selectivity (see Appendix A.3). However, they also observed a correlation between the number of concept-selective units and performance on the action40 dataset across networks and architectures. It is difficult to compare these results directly, as the data used are substantially different as is the method of evaluating selectivity. Nevertheless, we note that Bau et al. (2017) measured the absolute number of concept-selective units across networks with different total numbers of units and depths. The relationship between the number of concept-selective units and network performance may therefore arise as a result of a larger number of total units (if a fixed fraction of units is concept-selective) and increased depth (we both observed that selectivity increases with depth).

## 5 DISCUSSION AND FUTURE WORK

In this work, we have taken an empirical approach to understand what differentiates neural networks which generalize from those which do not. Our experiments demonstrate that generalization capability is related to a network's reliance on single directions, both in networks trained on corrupted and uncorrupted data, and over the course of training for a single network. They also show that batch normalization, a highly successful regularizer, seems to implicitly discourage reliance on single directions.

One clear extension of this work is to use these observations to construct a regularizer which more directly penalizes reliance on single directions. As it happens, the most obvious candidate to regularize single direction reliance is dropout (or its variants), which, as we have shown, does not appear to regularize for single direction reliance past the dropout fraction used in training (Section 3.3). Interestingly, these results suggest that one is able to predict a network's generalization performance without inspecting a held-out validation or test set. This observation could be used in several interesting ways. First, in situations where labeled training data is sparse, testing networks' reliance on single directions may provide a mechanism to assess generalization performance without sacrificing training data to be used as a validation set. Second, by using computationally cheap empirical measures of single direction reliance, such as evaluating performance at a single ablation point or sparsely sampling the ablation curve, this metric could be used as a signal for early-stopping or

hyperparameter selection. We have shown that this metric is viable in simple datasets (Section 3.2), but further work will be necessary to evaluate viability in more complicated datasets.

Another interesting direction for further research would be to evaluate the relationship between single direction reliance and generalization performance across different generalization regimes. In this work, we evaluate generalization in which train and test data are drawn from the same distribution, but a more stringent form of generalization is one in which the test set is drawn from a unique, but overlapping distribution with the train set. The extent to which single direction reliance depends on the overlap between the train and test distributions is also worth exploring in future research.

This work makes a potentially surprising observation about the role of individually selective units in DNNs. We found not only that the class selectivity of single directions is largely uncorrelated with their ultimate importance to the network's output, but also that batch normalization decreases the class selectivity of individual feature maps. This result suggests that highly class selective units may actually be *harmful* to network performance. In addition, it implies than methods for understanding neural networks based on analyzing highly selective single units, or finding optimal inputs for single units, such as activation maximization (Erhan et al., 2009) may be misleading. Importantly, as we have not measured feature selectivity, it is unclear whether these results will generalize to feature-selective directions. Further work will be necessary to clarify all of these points.

## ACKNOWLEDGMENTS

We would like to thank Chiyuan Zhang, Ben Poole, Sam Ritter, Avraham Ruderman, and Adam Santoro for critical feedback and helpful discussions.

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

# A  APPENDIX

## A.1  COMPARISON OF ABLATION METHODS

To remove the influence of a given direction, its value should be fixed or otherwise modified such that it is no longer dependent on the input. However, the choice of such a fixed value can have a substantial impact. For example, if its value were clamped to one which is highly unlikely given its distribution of activations across the training set, network performance would likely suffer drastically. Here, we compare two methods for ablating directions: ablating to zero and ablating to the empirical mean over the training set. Using convolutional networks trained on CIFAR-10, we performed cumulative ablations, either ablating to zero or to the feature map's mean (means were calculated independently for each element of the feature map), and found that ablations to zero were significantly less damaging than ablations to the feature map's mean (Fig. A1). Interestingly, this corresponds to the ablation strategies generally used in the model pruning literature (Li et al., 2017, Anwar et al., 2015, Molchanov et al., 2017).

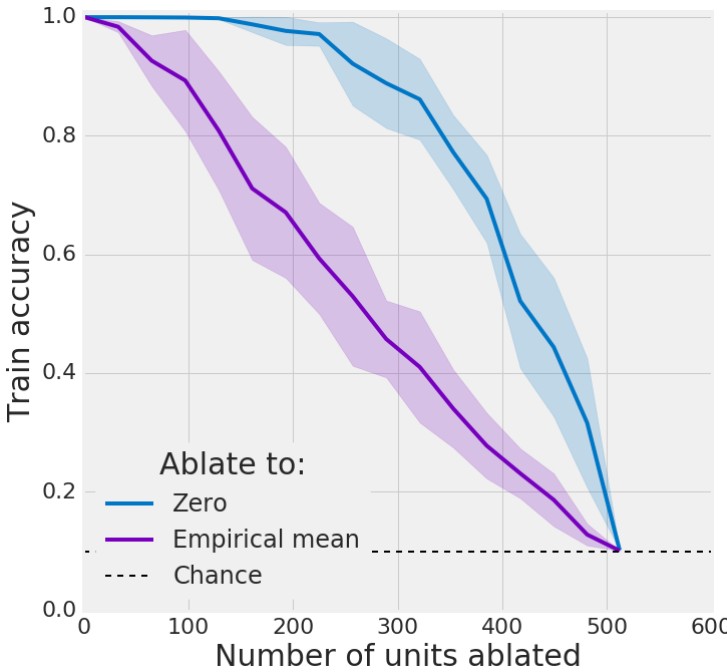

**Figure A1:  Ablation to zero vs. ablation to the empirical feature map mean.**

## A.2  TRAINING DETAILS

**MNIST MLPs**  For class selectivity, generalization, early stopping, and dropout experiments, each layer contained 128, 512, 2048 and 2048 units, respectively. All networks were trained for 640 epochs, with the exception of dropout networks which were trained for 5000 epochs.

**CIFAR-10 ConvNets**  Convolutional networks were all trained on CIFAR-10 for 100 epochs. Layer sizes were: 64, 64, 128, 128, 128, 256, 256, 256, 512, 512, 512, with strides of 1, 1, 2, 1, 1, 2, 1, 1, 2, 1, 1, respectively. All kernels were 3x3. For the hyperparameter sweep used in Section 3.2, learning rate and batch size were evaluated using a grid search.

**ImageNet ResNet**  50-layer residual networks (He et al., 2015) were trained on ImageNet using distributed training with 32 workers and a batch size of 32 for 200,000 steps. Blocks were structured as follows (stride, filter sizes, output channels): (1x1, 64, 64, 256) x 2, (2x2, 64, 64, 256), (1x1, 128, 128, 512) x 3, (2x2, 128, 128, 512), (1x1, 256, 256, 1024) x 5, (2x2, 256, 256, 1024), (1x1, 512, 512, 2048) x 3. For training with partially corrupted labels, we did not use any data augmen-

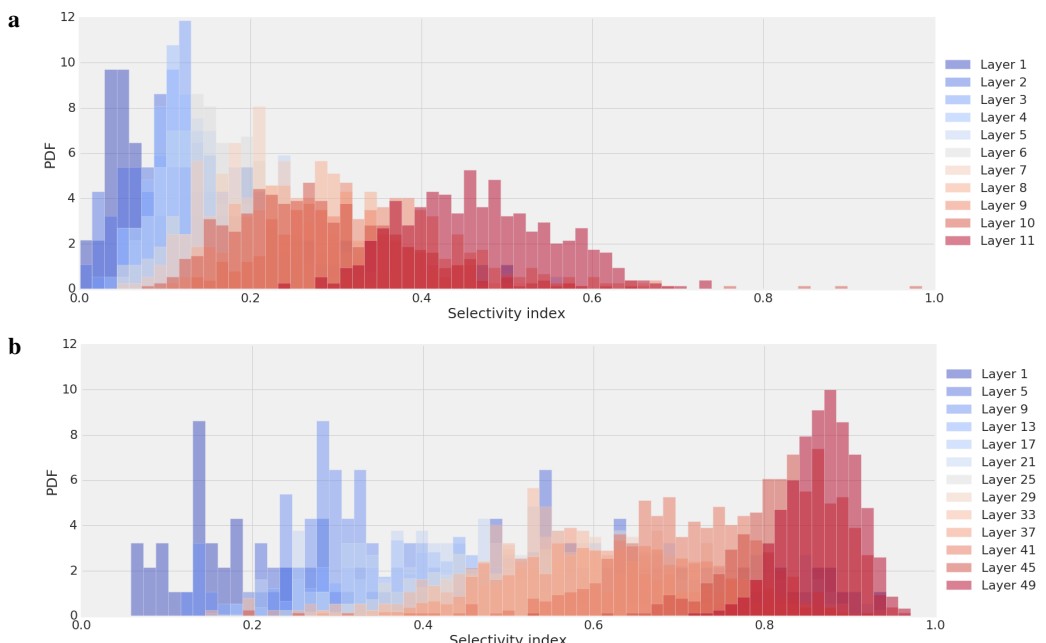

**Figure A2: Class selectivity increases with depth.** Class selectivity distributions as a function of depth for CIFAR-10 (**a**) and ImageNet (**b**).

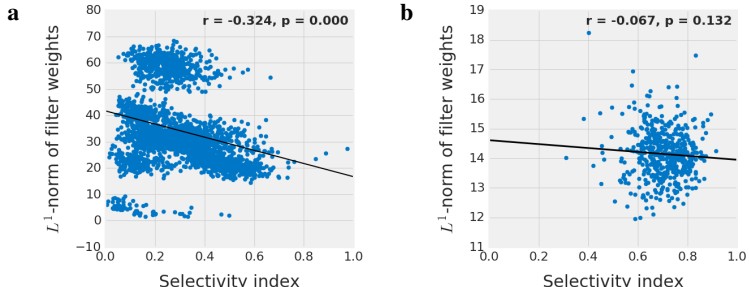

**Figure A3: Class selectivity is uncorrelated with $L^1$-norm.** Relationship between class selectivity and the $L^1$-norm of the filter weights for CIFAR-10 (**a**) and ImageNet (**b**).

tation, as it would have dramatically increasing the effective training set size, and hence prevented memorization.

### A.3 DEPTH-DEPENDENCE OF CLASS SELECTIVITY

Here, we evaluate the distribution of class selectivity as a function of depth. In both networks trained on CIFAR-10 (Fig. A2a) and ImageNet (Fig. A2b), selectivity increased as a function of depth. This result is consistent with Bau et al. (2017), who show that concept-selectivity increases with depth. It is also consistent with Alain & Bengio (2016), who show depth increases the linear decodability of class information (though they evaluate linear decodability based on an entire layer rather than a single unit).

### A.4 RELATIONSHIP BETWEEN CLASS SELECTIVITY AND THE FILTER WEIGHT $L^1$-NORM

Importantly, our results on the lack of relationship between class selectivity and importance do not suggest that there are not directions which are more or less important to the network's output, nor do they suggest that these directions are not predictable; they merely suggest that class selectivity is not a good predictor of importance. As a final test of this, we compared class selectivity to the

$L^1$-norm of the filter weights, a metric which has been found to be a strongly correlated with the impact of removing a filter in the model pruning literature (Li et al., 2017). Since the $L^1$-norm of the filter weights is predictive of impact of a feature map's removal, if class selectivity is also a good predictor, the two metrics should be correlated. In the ImageNet network, we found that there was no correlation between the $L^1$-norm of the filter weights and the class selectivity (Fig. A3a), while in the CIFAR-10 network, we found there was actually a negative correlation (Fig. A3b).

## A.5 RELATIONSHIP BETWEEN MUTUAL INFORMATION AND IMPORTANCE

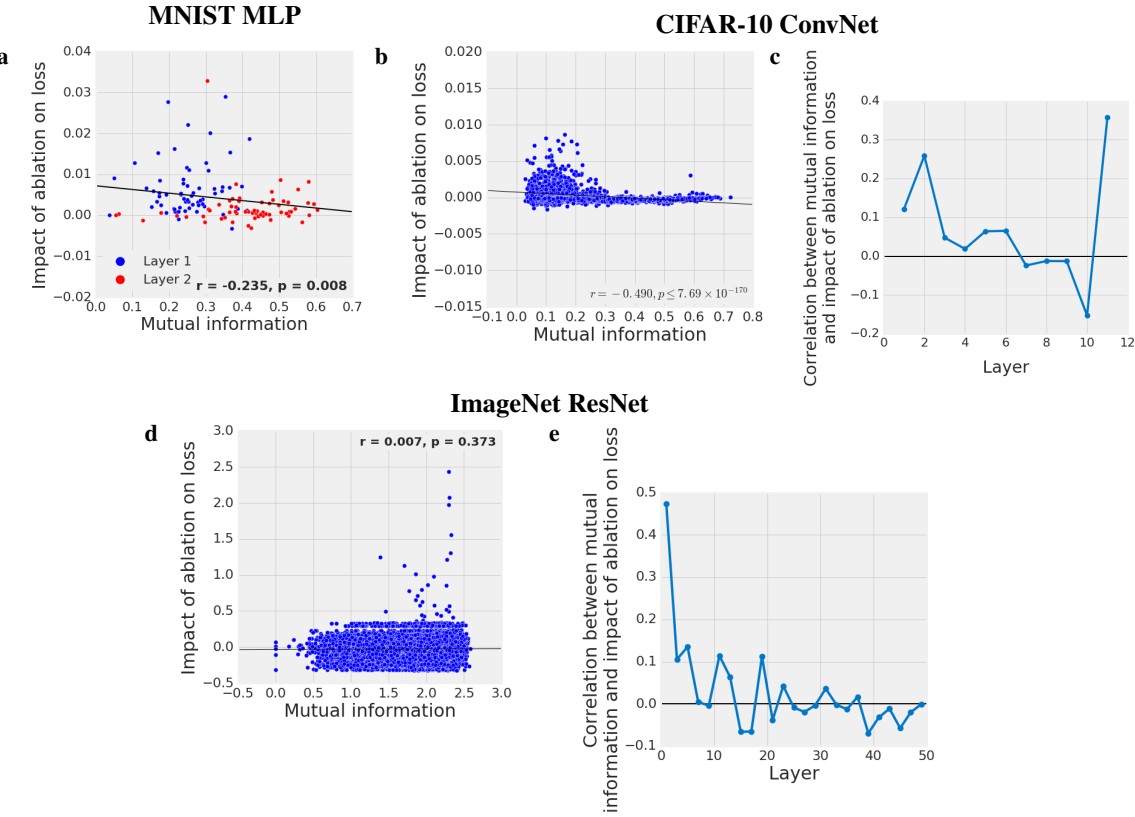

**Figure A4: Mutual information is not a good predictor of unit importance.** Impact of ablation as a function of mutual information for MNIST MLP (**a**), CIFAR-10 convolutional network (**b-c**), and ImageNet ResNet (**d-e**). **c** and **e** show regression lines for each layer separately.

To examine whether mutual information, which, in contrast to class selectivity, highlights units with information about multiple classes, is a good predictor of importance, we performed the same experiments as in Section 3.4 with mutual information in place of class selectivity. We found, that while the results were a little less consistent (e.g., there appears to be some relationship in very early and very late layers in CIFAR-10), mutual information was generally a poor predictor of unit importance (Fig. A4).

