# OpenReview forum: "On the importance of single directions for generalization"
_ICLR.cc/2018/Conference — Accept (Poster)_

### Official Review · AnonReviewer1 · 2017-11-26

**Rating:** 7
**Confidence:** 3

**Review:**


Summary:
- nets that rely on single directions are probably overfitting
- batch norm helps not having large single directions
- high class selectivity of single units is a bad measure to find "important" neurons that help a NN generalize.

The experiments that this paper does are quite interesting, somewhat confirming intuitions that the community had, and bringing new insights into generalization. The presentation is good overall, but many minor improvements could help with readability.


Remarks:
- The first thing you should say in this paper is what you mean by "single direction", at least an intuition, to be refined later. The second sentence of section 2 could easily be plugged in your abstract.
- You should already mention in section 2.1 that you are using ReLUs, otherwise clamping to 0 might take a different sense.
- considering the lack of page limit at ICLR, making *all* your figures bigger would be beneficial to readability.
- Figure 2's y values drop rapidly as a function of x, maybe make x have a log scale or something that zooms in near 0 would help readability.
- Figure 3b's discrete regimes is very weird, did you actually look at how much these clusters converged to the same solution in parameter space?
- Figure 4a is nice, but an additional figure zooming in on the first 2 epochs would be really great, because that AUC curve goes up really fast in the beginning.
- Arpit et al. find that there is more cross-class information being shared for true labels than random labels. Considering you find that low class selectivity is an indicator of good generalization, would it make sense to look at "cross-class selectivity"? If a neuron learns a feature shared by 2 or more classes, then it has this interesting property of offering a discrimination potential for multiple classes at the same time, rather than just 1, making it more "useful" potentially, maybe less adversary prone?
- You say in the figure captions that you use random orderings of the features to perform ablation, but nowhere in the main text (which would be nice).

---

> ### Author Response · Authors · 2017-12-22
> **Response to AnonReviewer1**
>
> We thank the reviewer for their kind comments and helpful feedback. We have incorporated the reviewer’s suggestions into the manuscript, and feel that they have substantially improved the clarity of the work. We have also performed additional experiments to address the importance of units with information about multiple classes, as the reviewer suggests. Details of these changes are below:
>
> "The first thing you should say in this paper is what you mean by 'single direction.'"
>
> We have now defined ‘single directions’ in the abstract as the reviewer suggests, as well as adding an additional definition in the Introduction. We agree that this improves the clarity of the paper substantially.
>
> "You should already mention in section 2.1 that you are using ReLUs."
>
> We have moved section 2.3 to section 2.1, thereby highlighting that we are using ReLU’s from the start, as the reviewer suggests.
>
> "considering the lack of page limit at ICLR, making *all* your figures bigger would be beneficial to readability."
>
> We were perhaps overly concerned about the ICLR 8-page soft limit in the first draft. We have increased the size of all the figures, as the reviewer suggests, and indeed, this improves the presentation of the paper.
>
> "Figure 2's y values drop rapidly as a function of x, maybe make x have a log scale or something that zooms in near 0 would help readability."
>
> We have now re-plotted Figure 2 using a log scale for the x-axis. We feel it has substantially improved the figure. We thank the reviewer for the great suggestion!
>
> "Figure 3b's discrete regimes is very weird, did you actually look at how much these clusters converged to the same solution in parameter space?"
>
> We absolutely agree that these discrete regimes are very weird, and fully intend to chase down the cause, and more generally, evaluate empirical convergence properties of multiple networks with the same topology but different random seeds in future work. However, an initial investigation into the causes of these regimes suggests that the answer is not obvious, and we believe that this question is beyond the scope of the present work.
>
> "Arpit et al. find that there is more cross-class information being shared for true labels than random labels. Considering you find that low class selectivity is an indicator of good generalization, would it make sense to look at "cross-class selectivity"? If a neuron learns a feature shared by 2 or more classes, then it has this interesting property of offering a discrimination potential for multiple classes at the same time, rather than just 1, making it more "useful" potentially, maybe less adversary prone?"
>
> We agree with the reviewer, and indeed, we had included a discussion of the downsides of class selectivity in section titled ‘Quantifying class selectivity’. While class selectivity absolutely ignores units with information about multiple classes, it has been used extensively in neuroscience to find neurons with strong tuning properties (e.g., the cat neurons prominently featured in previous deep learning analyses). In contrast, a metric such as mutual information should highlight units that are informative about multiple classes (with ‘cross-class selectivity’), but not necessarily units that are obviously interpretable.
>
> However, we agree that it would be worthwhile to assess the relationship between cross-class selectivity (as measured by mutual information) and importance. To this end, we have performed a series of additional experiments using mutual information (Fig. 6b; A4; Section A.5). We found that while mutual information was slightly more predictive of unit importance than class selectivity it is still not a good predictor of unit importance (Fig. A4, p.15). Interestingly, while we had previously shown that batch normalization decreases class selectivity, we found that batch normalization actually increases mutual information (Fig. 6b, p.7). This result suggests that batch normalization encourages representations that are distributed across units as opposed to representations in which information about single classes is concentrated in single units. We have added text discussing these results in sections 2.3 (p.3) and 3.4 (p.7).
>
> "You say in the figure captions that you use random orderings of the features to perform ablation, but nowhere in the main text (which would be nice)."
>
> We have now included a statement in the main text saying that each ablation curve contains multiple random orderings (p.4, first incomplete paragraph).

---

### Official Review · AnonReviewer2 · 2017-11-26
**review of "On the importance of single directions for generalization"**

**Rating:** 5
**Confidence:** 4

**Review:**

This is an "analyze why" style of paper:  the authors attempt to explain the relationship between some network property (in this case, "reliance on single directions"), and a desired performance metric (in this case, generalization ability).   The authors quantify a variety of related ways to measure "reliance on single directions" and show that the more reliant on a single directions a given network is, the less well it generalizes.

Clarity:  The paper is fairly clearly written.  Sometimes key details are in the footnotes (e.g. see footnote 3) -- not sure why -- but on the  whole, I think the followed the paper reasonably well.

Quality: The work makes a good-faith attempt to be fairly systematic -- e.g evaluating several different types of network structures, with reasonable numbers of random initializations, and also illustrates the main point in several different comparatively independent-seeming ways.  I feel fairly confident that the results are basically right within the somewhat limited domain that the authors explore.

Originality: This work is one in a series of papers about the topic of trying to understand what leads to good generalization in deep neural networks. I don't know that the concept of "reliance on a single direction" seems especially novel to me, but on the other hand, I can't think of another paper that precisely investigates this notion the way it is done here.

Significance: The work touches on some important issues.  I think the demonstration that the existence of strongly class-selective neurons is not a good correlate for generalization is interesting.   This point illustrates something that has made me a bit uncomfortable with the trend toward "interpretable machine learning" that has been arising recently:  in many of those results, it is shown that some fraction of the units at various levels of a trained deepnet have optimal driving stimuli that seem somewhat interpretable, with the implication that the existence of such units is an important correlate of network performance.  There has even been some claims that better-performing networks have more "single-direction" interpretable units [1].  The fact that the current results seem directly in contradiction to that line of work is interesting, and the connections to batch normalization and dropout are for the same reason interesting.  However, I wish the authors had grappled more directly with the apparent contradiction with (e.g.) [1].   There is probably a kind of tradeoff here.   The closer the training dataset is to what is being tested for "generalization", the more likely that having single-direction units is useful; and vice-versa.   I guess the big question is: what types of generalization are actually demanded / desired in real deployed machine learning systems (or in the brain)?  How does those cases compare with the toy examples analyzed here?   The paper doesn't go far enough in really addressing these questions, but it is sort of beginning to make an effort.

However, for me the main failing of the paper is that it's fairly descriptive without being that prescriptive. Does using their metric of reliance on a single direction, as a regularizer in and of itself, add anything above any beyond existing regularizers (e.g. batch normalization or dropout)?  It doesn't seem like they tried. This seems to me the key question to understanding the significance of their results.   Is "reliance on single direction" actually a good regularizer as such, especially for "real" problems like (e.g.) training a deep Convnet on (e.g.) ImageNet or some other challenging dataset?  Would penalizing for this quantity improve the generalization of a network trained on ImageNet to other visual datasets (e.g. MS-COCO)?  If so, this would be a very significant result and would make me really care about their idea of "reliance on a singe direction".  If such results do not hold, it seems to me like one more theoretical possibility that would bite the dust when tested at scale.

[1] http://netdissect.csail.mit.edu/final-network-dissection.pdf

---

> ### Author Response · Authors · 2017-12-22
> **Response to AnonReviewer2 Part 1/2**
>
> We thank the reviewer for their constructive feedback and their thorough reading of our paper. We have performed additional experiments (to show that the insights of this work can be used prescriptively) and provided additional discussion to work towards addressing the concerns the reviewer has raised. We have provided detailed responses to these comments as well as pointers to changes in the paper below:
>
> "Sometimes key details are in the footnotes..."
>
> We initially put these details in footnotes to stay below the soft page limit. We have now moved all footnotes containing key details into the main text as the reviewer has requested.
>
> "Originality: This work is one in a series of papers about the topic of trying to understand what leads to good generalization in deep neural networks. I don't know that the concept of "reliance on a single direction" seems especially novel to me, but on the other hand, I can't think of another paper that precisely investigates this notion the way it is done here."
>
> As we discuss in both the introduction and related work sections of our paper, the concept of single direction reliance is related to previous theoretical work such as flat minima. However, to our knowledge, single direction reliance has never been empirically tested explicitly. Nonetheless, if the reviewer would be willing to point us in the direction of any related papers that we may have omitted from our manuscript, we would greatly appreciate it as we want to ensure that our discussion of prior work is as complete as possible.
>
> "There has even been some claims that better-performing networks have more "single-direction" interpretable units [1].  The fact that the current results seem directly in contradiction to that line of work is interesting, and the connections to batch normalization and dropout are for the same reason interesting.  However, I wish the authors had grappled more directly with the apparent contradiction with (e.g.) [1]."
>
> We have included an additional paragraph in the related work section (Section 4, p.9, third complete paragraph) comparing our work more extensively to the work of Bau et al. [1].  We believe that Bau et al. is extremely interesting work, and we note that, in many cases, our results are largely consistent with what Bau et al. observed; for example, we both found a relationship between selectivity and depth. However, we do acknowledge that they observed a correlation between network performance and the number of concept-selective units (Fig. 12 in Bau et al.). We believe that there are three potential explanations for this discrepancy:
>
>     (1) As we note at the end of Section 2.3, class selectivity and feature selectivity (akin to the concept selectivity used in Bau et al.) may exhibit different properties.
>
>     (2) Bau et al. compare networks with different numbers of filters (e.g., AlexNet, GoogleNet, VGG, and ResNet-152s), but measure the absolute number of unique detectors. It is possible that the number of unique detectors in better performing networks, such as ResNets, is simply a function of these networks having more filters.
>
>     (3) Finally, both Bau et al. and our work observed a relationship between selectivity and depth (see Fig. 5 in Bau et al., and Fig. A2 in our manuscript). As Bau et al. compared the number of unique detectors across networks with substantially different depths, the increase in the number of unique detectors may have been due to the different depths of these networks. In line with this observation (as well as point 2 above), we note that in Fig. 12 in Bau et al., which plots the number of unique detectors as a function of accuracy on the action40 dataset, there appears to be little relationship when comparing only across points from the same model architecture.
>
> "‘The closer the training dataset is to what is being tested for "generalization", the more likely that having single-direction units is useful; and vice-versa.   I guess the big question is: what types of generalization are actually demanded / desired in real deployed machine learning systems (or in the brain)?"
>
> We have now included an additional paragraph in the Discussion section (p.9 last incomplete paragraph) addressing the distinction between different types of generalization based on the overlap between the train and test distributions. We believe that understanding how single direction reliance varies based on this overlap is an extremely interesting question although we feel it is beyond the scope of the present work.
>
> [1] David Bau, Bolei Zhou, Aditya Khosla, Aude Oliva, and Antonio Torralba. Network Dissection: Quantifying Interpretability of Deep Visual Representations. 2017. doi: 10.1109/CVPR.2017. 354. URL http://arxiv.org/abs/1704.05796.

---

> > ### Author Response · Authors · 2017-12-22
> > **Response to AnonReviewer2 Part 2/2**
> >
> > "However, for me the main failing of the paper is that it's fairly descriptive without being that prescriptive. Does using their metric of reliance on a single direction, as a regularizer in and of itself, add anything above and beyond existing regularizers (e.g. batch normalization or dropout)?"
> >
> > Though we would like to note that the primary goal of this work is to understand what factors lead to good generalization performance rather than to engineer a new model, we agree with the reviewer that a demonstration that the insights from our work can be used to directly improve model performance would be extremely valuable. However, all of the most obvious methods to regularize single direction reliance seem to reduce to dropout or one of its close variants. This is not to say that we believe there is no such regularizer -- it is merely to say that it is not obviously apparent. We have added a sentence in the Discussion to this effect (p.9, last complete paragraph).
> >
> > Nonetheless, we do note that the insights from our work can be used prescriptively to indirectly improve models, as they provide a way to assess generalization performance without the need for a held-out validation set. In the original draft, we explored this in Fig. 4a-b as a means for early stopping. To expand on the potential for the method in this direction, we have added an additional experiment, in which we show that single direction reliance can be used as an effective method for hyperparameter selection as well (Fig. 4c, p.5 last complete paragraph). We believe that this approach may prove extremely useful, especially in situations in which labeled data is rare.

---

### Official Review · AnonReviewer3 · 2017-11-27
**An important piece of the generalization puzzle**

**Rating:** 9
**Confidence:** 3

**Review:**

article summary:
The authors use ablation analyses to evaluate the reliance on single coordinate-aligned directions in activation space (i.e. the activation of single units or feature maps) as a function of memorization. They find that the performance of networks that memorize more are also more affected by ablations. This result holds even for identical networks trained on identical data. The dynamics of this reliance on single directions suggest that it could be used as a criterion for early stopping. The authors discuss this observation in relation to dropout and batch normalization. Although dropout is an effective regularizer to prevent memorization of random labels, it does not prevent over-reliance on single directions. Batch normalization does appear to reduce the reliance on single directions, providing an alternative explanation for the effectiveness of batch normalization. Networks trained without batch normalization also demonstrated a significantly higher amount of class selectivity in individual units compared to networks trained without batch normalization. Highly selective units were found to be no more important than units that were not selective to a particular class. These results suggest that highly selective units may actually be harmful to network performance.

* Quality: The paper presents thorough and careful empirical analyses to support their claims.
* Clarity: The paper is very clear and well-organized. Sufficient detail is provided to reproduce the results.
* Originality: This work is one of many recent papers trying to understand generalization in deep networks. Their description of the activation space of networks that generalize compared to those that memorize is novel. The authors throughly relate their findings to related work on generalization, regularization, and pruning. However, the authors may wish to relate their findings to recent reports in neuroscience observing similar phenomena (see below).
* Significance: The paper provides valuable insight that helps to relate existing theories about generalization in deep networks. The insights of this paper will have a large impact on regularization, early stopping, generalization, and methods used to explain neural networks.

Pros:
* Observations are replicated for several network architectures and datasets.
* Observations are very clearly contextualized with respect to several active areas of deep learning research.
Cons:
* The class selectivity measure does not capture all class-related information that a unit may pass on.

Comments:
* Regarding the class selectivity of single units, there is a growing body of literature in neurophysiology and neuroimaging describing similar observations where the interpretation has been that a primary role of any neural pathway is to “denoise” or cancel out the “distractor” rather than just amplifying the “signal” of interest.
    * Untuned But Not Irrelevant: The Role of Untuned Neurons In Sensory Information Coding, https://www.biorxiv.org/content/early/2017/09/21/134379
    * Correlated variability modifies working memory fidelity in primate prefrontal neuronal ensembles https://www.ncbi.nlm.nih.gov/pubmed/28275096
    * On the interpretation of weight vectors of linear models in multivariate neuroimaging http://www.sciencedirect.com/science/article/pii/S1053811913010914
        * see also LEARNING HOW TO EXPLAIN NEURAL NETWORKS https://openreview.net/forum?id=Hkn7CBaTW
* Regarding the intuition in section 3.1, "The minimal description length of the model should be larger for the memorizing network than for the structure- finding network. As a result, the memorizing network should use more of its capacity than the structure-finding network, and by extension, more single directions”. Does reliance on single directions not also imply a local encoding scheme? We know that for a fixed number of units, a distributed representation will be able to encode a larger number of unique items than a local one. Therefore if this behaviour was the result of needing to use up more of the capacity of the network, wouldn’t you expect to observe more distributed representations?

Minor issues:
* In the first sentence of section 2.3, you say you analyzed three models and then you only list two. It seems you forgot to include ResNet trained on ImageNet.

---

> ### Author Response · Authors · 2017-12-22
> **Response to AnonReviewer3**
>
> First off, we would like to thank the reviewer for the kind review and the helpful feedback, especially with respect to class selectivity and the relationship to neuroscience. We have provided detailed responses to these comments as well as pointers to changes in the paper below:
>
> "The class selectivity measure does not capture all class-related information that a unit may pass on."
>
> We agree with the reviewer, and indeed, we had included a discussion of the downsides of class selectivity in section titled ‘Quantifying class selectivity.’ While class selectivity absolutely ignores units with information about multiple classes, it has been used extensively in neuroscience to find neurons with strong tuning properties (e.g., the cat neurons prominently featured in previous deep learning analyses). In contrast, a metric such as mutual information should highlight units that are informative about multiple classes, but not necessarily units that are obviously interpretable.
>
> However, we agree that it would be worthwhile to assess the relationship between multi-class selectivity (as measured by mutual information) and importance. To this end, we have performed a series of additional experiments using mutual information (Fig. 6b; A4; Section A.5). We found that while mutual information was slightly more predictive of unit importance than class selectivity, it is still not a good predictor of unit importance (Fig. A4, p.15). Interestingly, while we had previously shown that batch normalization decreases class selectivity, we found that batch normalization actually increases mutual information (Fig. 6b, p.7). This result suggests that batch normalization encourages representations that are distributed across units as opposed to representations in which information about single classes is concentrated in single units. We have added text discussing these results in sections 2.3 (p.3) and 3.4 (p.7).
>
> "... the authors may wish to relate their findings to recent reports in neuroscience ..."
>
> We are strong advocates of the idea that methods and ideas from neuroscience are useful for understanding machine learning models, and so, we have also included an additional paragraph in our ‘related work’ section (p.8, first complete paragraph) contextualizing our work in recent neuroscience developments regarding robustness to noise, distributed representations, and correlated variability, including references that the reviewer has provided and several other neuroscience papers that influenced our work.
>
> "In the first sentence of section 2.3, you say you analyzed three models and then you only list two. It seems you forgot to include ResNet trained on ImageNet."
>
> Great catch! We have resolved this now.

---

### Author Response · Authors · 2017-12-22
**General response to reviewers accompanying revision**

We wish to thank the reviewers for their thoughtful and thorough reviews. In particular, we are glad that the reviewers found our paper to be "an important piece of the generalization puzzle," that "the work touches on some important issues," and that the experiments are "quite interesting," "bringing new insights into generalization." We are also glad that the reviewers found that the paper contains "thorough and careful empirical analyses," "is very clear and well-organized," and that the "presentation is good overall."

To address the reviewers' comments we have performed several additional experiments, including additional figures expanding on the prescriptive implications of our work and detailing the relationship between mutual information, batch normalization, and unit importance.  We have also made a number of changes to the text which we feel have significantly improved its clarity. For detailed descriptions of the changes we have made, please see our responses to individual reviewers below. As a result of these changes, our paper is now a little more than nine pages long. Due in large part to the reviewers’ constructive feedback, we believe that our paper has been substantially strengthened.

---

### Public Comment · ~Tom_Diethe1 · 2018-04-03
**Code**

Do you plan to release a repository to replicate the experiments?

---

### Decision · Program_Chairs · 2018-01-29
**ICLR 2018 Conference Acceptance Decision**

**Decision:**

Accept (Poster)

**Comment:**

The paper contributes to a body of empirical work towards understanding generalization in deep learning. They do this  through a battery of experiments studying "single directions" or selectivity of small groups of neurons. The reviewers that have actively participated agree that the revision is of high quality, impact, originality, and significance. The issue of a lack of prescriptiveness was raised by one reviewer. I agree with the majority that this is not necessary, but nevertheless, the revision makes some suggestions.  I urge the authors to express the appropriate amount of uncertainty regarding any prescriptions that have not been as thoroughly vetted!